# SNHG15-Mediated Localization of Nucleolin at the Cell Protrusions Regulates CDH2 mRNA Expression and Cell Invasion

**DOI:** 10.3390/ijms242115600

**Published:** 2023-10-26

**Authors:** Shaoying Chen, Yanchun Zhou, Pei Peng, Liqun Xu, Quandong Tang, Weibin Chen, Wei Gu

**Affiliations:** Key Immunopathology Laboratory of Guangdong Province, Department of Pathophysiology, Shantou University Medical College, Shantou 515041, China; sychen3@stu.edu.cn (S.C.); zyc2013st@foxmail.com (Y.Z.); peng23pei@163.com (P.P.); xuliqun@hb-biopharma.com (L.X.); 19qdtang@stu.edu.cn (Q.T.); wbcu@stu.edu.cn (W.C.)

**Keywords:** lncRNA SNHG15, nucleolin, N-cadherin (CDH2), RNA localization, protein interaction, post-transcriptional regulation

## Abstract

LncRNAs are emerging as important regulators of gene expression by controlling transcription in the nucleus and by modulating mRNA translation in the cytoplasm. In this study, we reveal a novel function of lncRNA SNHG15 in mediating breast cancer cell invasion through regulating the local translation of CDH2 mRNA. We show that SNHG15 preferentially localizes at the cellular protrusions or cell leading edge and that this localization is directed by IMP1, a multifunctional protein involved in many aspects of RNA regulation. We demonstrate that SNHG15 also forms a complex with nucleolin, allowing nucleolin to be co-transported with SNHG15 to the cell protrusions, where the accumulated nucleolin is able to bind to CDH2 mRNA. Interaction with nucleolin stabilizes local CDH2 mRNA and regulates its translation, thus promoting cell invasive potential. Our findings reveal an underlying mechanism by which lncRNA could serve as a carrier to transport a protein regulator into a specific cell compartment to enhance target mRNA expression.

## 1. Introduction

Breast cancer is the most frequent malignancy in females around the world, and twenty percent of cases develop metastasis [1]. Since breast cancer metastasis is usually resistant to conventional therapy and provides only a small chance for successful treatment, identification of new molecular markers that affect cell invasion and investigation of the underlying mechanisms related to changes in cell migration and adhesion are urgently required [2].

Eukaryote whole-genome sequencing revealed that only 2% of genes are related to protein coding [3]. Transcription of the entire eukaryotic genome generates huge amounts of non-coding RNA species, among which lncRNAs (long non-coding RNAs) with over 200 nucleotides form a subset of ncRNAs [4,5]. Recently, growing evidence has indicated that lncRNAs can function as promising oncogenes or tumor-suppressor genes to participate in various biological processes, including cell proliferation, cell cycle arrest, anti-apoptosis and metastasis [6,7,8,9]. Until now, a few lncRNAs have been well studied, such as metastasis-associated lung adenocarcinoma transcript-1 (MALAT1) [10], urothelial carcinoma-associated 1 (UCA1) [11] and H19 lncRNA [12]. The functions of most lncRNAs still remain unknown and need to be further investigated.

It has been widely considered that lncRNA could regulate gene expression at the epigenetic, transcriptional and translational levels, and these regulations often occur through interactions with RNA-binding proteins (RBPs) [13,14]. Correspondingly, RBPs are also able to bind to a heterogeneous class of functional lncRNAs to modulate their biological effects [15]. Protein–RNA interactions are important aspects of many cellular processes that go beyond the already recognized steps of mRNA regulation, including RNA stability, localization and translation [16,17]. IMP1 (insulin-like growth factor 2 mRNA binding protein) is one of those RBPs that can also mediate the activity of lncRNAs [18,19]. Previous studies showed that IMP1 not only mediates the localized translation of a group of mRNAs at the leading edge [20,21] but is also involved in the post-transcriptional regulation of lncRNAs in tumor cells [19,22]. The loss of IMP1 function deregulates many RNAs normally associated with the protein, resulting in increased cell invasive ability [21,23].

SNHG15 (small nucleolar RNA host gene 15), located on chromosome 7p13, was initially identified as a substitute indicator for chemical-induced stress responses [24]. Recent studies have suggested that SNHG15 is involved in cancer occurrence and progression and is linked with poor survival in many human malignancies, including colorectal cancer, breast cancer, lung cancer, ovarian cancer and pancreatic cancer [25,26,27]. SNHG15 expression was increased in breast tumors and has been shown to be positively associated with larger tumor size and lymph-node metastasis. Mechanistic investigations indicate that SNHG15 could potentially act as a ceRNA (competing endogenous RNA) to sponge individual miRNAs, such as miR-211-3p [25], miR-451 [28] and miR-411-5p [27], to regulate cell invasion, apoptosis and EMT. Based on the role of SNHG15 in cancer progression, more investigation of the mechanisms by which SNHG15 mediates oncogenic processes is necessary to better understand its function and provide essential insight in the clinical treatment of cancer patients.

In this study, we report that in breast cancer cells, SNHG15 is one of the lncRNAs that interacts with IMP1. We identified that a conserved “ACACCC” motif within SNHG15 was responsible for IMP1 binding, and this binding enhanced SNHG15 to be localized at the cell protrusions or the leading edge. We demonstrated that SNHG15 could also form a complex with nucleolin, a multifunction protein with many features of RNA regulation, through which nucleolin was co-transported into the cell protrusions. SNHG15-mediated accumulation of nucleolin at the cell protrusions allowed it to interact with CDH2 mRNA and regulate its local translation, thus promoting cell invasive potential. Our study provides initial evidence that an lncRNA performs an oncogenic role through serving as a carrier to localize a particular protein to a particular cell compartment.

## 2. Results

### 2.1. SNHG15 Promotes the Proliferation and Invasion of Breast Cancer Cells

We have previously shown that, in breast cancer cells, IMP1 alters the expression of many lncRNAs and participates in their biological activities [19]. To investigate the potential mechanism by which IMP1 regulates the function of lncRNAs, we performed IMP1-RIP (RNA immunoprecipitation) followed by RNA-seq assays to identify lncRNAs that were associated with IMP1 in T47D cells. About 90 lncRNAs were identified in the IMP1 precipitates (the raw micro-seq data have been deposited in the Gene Expression Omnibus (GEO) database under submission number GSE220087). Some IMP1-associated lncRNAs involved in tumorigenesis are listed in Appendix A. To ensure that the association of IMP1 with these identified lncRNA is a common circumstance in breast cancer cells, we performed IMP1-RIP and RT-qPCR assays in a BT-549 cell line and selected six lncRNAs to test their association. All six lncRNAs, including SNHG15, were truly bound to IMP1 (Figure 1A). SNHG15 is not a well-studied oncogene. It localizes to human chromosome 7p13 and encodes five different non-coding transcripts with the sizes of 860 bp (V1), 783 bp (V2), 774 bp (V3), 983 bp (V4) and 713 bP (V5) in human tumors (https://www.ncbi.nlm.nih.gov/nuccore/NR_152596.1, accessed on 4 June 2023). RT-PCR showed that the 983 bp SNHG15 (V4) is the major transcript expressed in breast cancer cells (Figure 1B). Investigation of the endogenous levels of SNHG15 in four breast cancer cell lines indicated that T47D cells expressed the highest levels of SNHG15, while BT-549 cell lines expressed relatively low levels of the lncRNA (Figure 1C and Appendix A). Functional assays indicated that silencing SNHG15 in T47D and MDA-MB-231 cells decreased cell proliferation potential (Figure 1D and Appendix A), while ectopically expressed SNHG15 in BT-549 cells significantly promoted cell proliferation and invasive ability (Figure 1E), indicating the oncogenic role of SNHG15 in breast cancer cells.

### 2.2. IMP1 Regulates the Localization of SNHG15 at the Cell Protrusions

Since the cellular localization of biomolecules closely relates to their functions, we examined the subcellular localization of SNHG15. FISH (fluorescence in situ hybridization) experiments indicated that either endogenously or ectopically expressed SNHG15 in BT-549 cells localized at the cell protrusions or the leading edge (Figure 2A). Based on the facts that IMP1 can regulate the localization and translation of a plethora of mRNAs [18] and that IMP1 interacts with SNHG15 (Figure 1), we rationalized that the localization of SHNG15 was mediated by IMP1. To address this, we performed SNHG15 FISH in a pair of stable MDA-MB-231 cell lines with or without IMP1 expression. Since WT (wild-type) MDA-MB-231 cells expressed inferior levels of endogenous IMP1 [29], less than 40% of cell population displayed SNHG15 localization (Figure 2B, upper and Figure 2C). In comparison, the localization of SNHG15 at the cell protrusion was significantly increased to nearly 70% in the cells that ectopically expressed IMP1 (Figure 2B, lower and Figure 2C). Consistently, SNHG15 co-localized with IMP1 (Appendix A) in BT-549 cells, and knocking down IMP1 mRNA by siRNA apparently decreased SNHG15 localization (Appendix A). qRT-PCR showed that IMP1 expression barely affected cellular levels of SNHG15 (Figure 2D), suggesting that IMP1 plays a role in SNHG15 localization but not its expression.

### 2.3. Localization of Full-Length SNHG15 at the Cell Protrusions Increases the Potential of Cell Invasion

Since IMP1 mediated the protrusion localization of SNHG15, we searched the nucleotide sequence of SNHG15 lncRNA and identified a conserved motif of “ACACCC”, which was previously shown as a key sequence for IMP1 binding in β-actin mRNA [30], located in the region of 407–412 nt (Figure 3A and Appendix A). Deletion of the motif-containing region notably reduced the localization of SNHG15 at the cell protrusions by nearly 35% (Figure 3B,C). We next divided the full-length SNHG15 (983 nt) into three truncated segments, (T1: 1–480 bp; T2: 460–983 bp; T3: 251–700 bp). Each of them was tagged with six MS2 hairpin loop repeats (Figure 3A), which were recognized by a recombinant protein MBP–MCP and allowed us to perform RNA pull-down assays [19]. We then established BT-549 cell lines stably expressing SNHG15 truncates. After verifying the expression of each truncate by qRT-PCR (Appendix A), we performed FISH experiments, which showed that T1 and T3 truncates could localize at the cell protrusions, while the localization of the T2 truncate was largely decreased because it lacked the “ACACCC” localization motif (Figure 3D). Interestingly, in comparison with the full-length SHNG15, all three truncates, including T1 and T3, displayed lower invasive ability (Figure 3E). These results suggest that full-length SNHG15 is required for the oncogenic role of SNHG15 in breast cancer cells.

### 2.4. SNHG15 Forms a Complex with Nucleolin and Carries Nucleolin to the Cell Protrusions

RNA–protein interactions play important roles in mediating the activity of their binding partners. Since ectopic SNHG15 in stable BT-549 cells was tagged with six MS2 hairpin repeats, we used a recombinant MBP–MCP to pull down SNHG15 in BT-549 cells and analyzed its associated proteins by MS spectrometry (the data have been submitted to the ProtemeXchange database under submission number IPX0005630000). A few RNA-binding proteins were identified in the SNHG15 precipitates, including nucleolin (NCL), DDX5, PABP1 and FUS (Appendix A). Among these SNHG15-associated proteins, we were mostly interested in nucleolin, which is not only a nuclear protein but has also been reported to be an oncogenic factor by regulating particular cytoplasmic mRNAs [31]. In vivo binding of SNHG15 to nucleolin was further confirmed by NCL-RIP and RT-PCR (Figure 4A). Immuno-staining using nucleolin antibody showed that although nucleolin was mainly located in the nucleolus, a considerable amount of nucleolin was accumulated at the cell protrusions of BT-549 cells (Figure 4B,C), as well as in MDA-MB-231/IMP1 cells (Appendix A). Strongly accumulated nucleolin was observed in nearly 80% of BT-549 cells that ectopically expressed SNHG15, while the accumulation of nucleolin was seen only in 30% of WT BT-549 cells. Moreover, IF and FISH double staining assays demonstrated that nucleolin was co-localized with SHNG15 at the cell protrusions of both BT-549 and MDA-MB-231 cells (Figure 4D and Appendix A), indicating that SNHG15 serves as a carrier to bring nucleolin to the cell protrusions. Sequence analysis predicted an AT-rich fragment within the 3′end of SNHG15 (Appendix A), which has been reported for nucleolin binding [31]. RIP followed by RT-qPCR indicated that in contrast to the full-length SNHG15, truncated SNHG15 (T3) lacking this fragment lost the ability to bind to nucleolin (Appendix A). These data suggested that nucleolin was co-transported with SNHG15 to the cell protrusions through binding to the 3′end of SNHG15.

### 2.5. Expression of CDH2 mRNA Was Altered in Response to SNHG15 Expression

To investigate the biological impact of the SHNG15-mediated accumulation of nucleolin at the cell protrusions, we isolated total RNAs from BT-549 cells with or without ectopic SNHG15 expression and examined the differential expression of genes through mRNA-seq assays performed by Shanghai Biotechnology Corporation (China). Figure 5A shows a volcano plot with differentially expressed genes in BT549-SHNG15 cells compared with BT549-PCPI2 cells. A total of 1551 transcripts with at least a 2-fold change between the two cell lines were identified (data have been deposited in the Gene Expression Omnibus under submission number GSE220492), among which 929 genes were up-regulated and 622 genes were down-regulated in response to SNHG15 expression (Figure 5A, *p* < 0.01). Six significant individual transcripts, including two up-regulated mRNAs (CDH2 and TPD52) and four down-regulated mRNAs (TFPI2, MTSS1, MYCT1 and TIMP3) were selected (Appendix A) to confirm their differential expression in MDA-MB-231 cells. Consistently, all six selected mRNAs displayed a similar differential expression pattern in response to SHNG15 expression (Figure 5A). Since CDH2 plays a vital role in cancer cell invasion and EMT [32], we investigated whether increased levels of CDH2 mRNA in SNHG15-expressing cells resulted from the accumulation of nucleolin at the cell protrusions. FISH showed that although CDH2 mRNA was widely distributed in WT breast cancer cells, increased amounts of CDH2 mRNA were observed in cell protrusions when SNHG15 was overexpressed (Figure 5B). As a result, over 70% of cells showed stronger CDH2 protein signals at the protrusions of SNHG15-expressing cells than in WT cells (Figure 5C,D). In comparison, the expression of CDH1 (E-cadherin), which represses cell invasion, was not much affected by SNHG15 expression (Appendix A). Western blots followed by densitometric analyses also showed that levels of CDH2, but not CDH1, were elevated in cells expressing ectopic SNHG15 (Figure 5E). These results suggest that SNHG15-mediated nucleolin accumulation would enhance the local translation of CDH2 mRNA.

### 2.6. Accumulation of Nucleolin at the Cell Protrusion Upregulates Local Translation of CDH2 mRNA and Increases Cell Invasive Potential

We hypothesized that SNHG15-mediated nucleolin accumulation at the cell protrusions would post-transcriptionally regulate the local translation of CDH2 mRNA. To address this hypothesis, we performed nucleolin-RIP assays, which demonstrated that CDH2 mRNA was preferentially co-precipitated with nucleolin (Figure 6A,B). Consistently, IF and FISH double staining assays indicated that nucleolin was strongly co-localized with CDH2 mRNA at the protrusions of more than 70% of BT-549 cells, which expressed ectopic SNHG15 (Figure 6C,D). We then constructed a luciferase reporter in which the 3′UTR of CDH2 mRNA was fused into the 3′end of the renilla gene (Figure 6E, upper) and then transfected the reporter into stable SNHG15-expressing BT-549 cells or MDA-MB-231 cells in which SNHG15 was knocked down by siRNA. Luciferase activity was significantly higher in cells expressing SNHG15 (Figure 6E, lower), while the activity was lower in MDA-MB-231 cells when SNHG15 was silenced (Figure 6F and Appendix A). Finally, we performed Transwell assays, which showed that SHNG15-induced cell invasion could be rescued either by knocking down SNHG15 expression or by silencing CDH2 mRNA in BT-549/SNHG15 (Figure 6G,H). In MA-MB-231 cells, invasive potential could also be reduced by knocking down CDH2 mRNA (Appendix A). These results suggest that SHNG15-mediated enrichment of nucleolin at the cell protrusions increases the invasive potential of breast cancer cells through regulation of the local translation of CDH2 mRNA.

## 3. Discussion

Increasing evidence indicates that lncRNAs play vital roles in cancer progression. However, only a few lncRNAs have been functionally and mechanically characterized. In this study, we revealed a novel mechanism in which SHNG15-mediated carcinogenesis occurs at least in part through the interactions with IMP1 and nucleolin. Specifically, SNHG15 forms a complex with nucleolin and interacts with IMP1. This interaction allows IMP1 to localize the SNHG15/nucleolin complex at the cell leading edge where nucleolin is released. The accumulation of nucleolin at the cell leading edge or protrusions was able to interact with CDH2 mRNA to stabilize the mRNA and enhance its local translation to promote breast cancer cell invasion.

A large number of lncRNAs have been reported to function as competing endogenous RNA (ceRNA) [33]. These lncRNAs, including SNHG15, have been demonstrated to serve as molecular sponges for miRNAs and functionally regulate the activity of the RNA transcripts targeted by sponged miRNAs [25,27,28]. Our findings indicated that, in addition to acting as a ceRNA to sponge miRNAs, SNHG15 also interacts with different RNA-binding proteins, and through these interactions, SNHG15 can be localized and carry its associated partners to a specific cell compartment. In this process, IMP1, a well-studied RNA regulator involved in many aspects of RNA regulation [34], is responsible for SNHG15 localization, and nucleolin, another multifunctional RNA-binding protein, is a cargo to be co-transported to the cell protrusions. Truncated SNHG15 (T1, T2 and T3), which neither binds to IMP1 nor interacts with nucleolin, was unable to properly localize nucleolin to the cell protrusions and thus showed lower invasive capability (Figure 3, Figure 4 and Figure 5).

In breast cancer cells, cytoplasmic nucleolin influences cell survival, proliferation and invasion through its action on the post-transcriptional regulation of tumor-related mRNAs [35]. Currently, the effect of nucleolin in the progression of cancer has received much attention. Disordered accumulation of nucleolin was observed in various cancers and has been shown to play important roles during tumorigenesis [36]. Nucleolin affects the translation or turnover of many target mRNAs, such as MMP9 and BCL2 mRNAs, which usually bear AU-rich elements (AREs), typically present in their 3′UTR [37,38]. Interestingly, the 3′UTR of CDH2 mRNA also contains multiple AU-rich elements. It is most likely that the increased expression of CDH2 mRNA at the cell protrusions could be regulated by the SNHG15-mediated local enrichment of nucleolin.

CDH2 (N-cadherin) is a calcium-dependent transmembrane protein that mediates cell–cell adhesion. It is increasingly known that aberrant expression of CDH2 is closely related to malignant tumor processes such as apoptosis and metastasis and is thus considered a mesenchymal marker of the EMT phenotype [39]. In breast tumors, CDH2 expression is correlated with invasion through CDH2-mediated interactions between breast cancer cells and stromal cells [40]. An early study by Nieman et al. indicated that CDH2 promoted cell motility and invasion regardless CDH1 (E-cadherin) expression [41]. Recently, an increasing number of signaling pathways and regulatory factors, including β-catenin, NF-κB, EGFR and STAT3, have been found to regulate CDH2 gene transcription [42]. In addition, miR-145 was shown to inhibit the translation of CDH2 mRNA by directly binding to its 3′-untranslated region [43]. Our results demonstrated a novel mechanism for the post-transcriptional regulation of CDH2 mRNA, in which the SNHG15-mediated accumulation of nucleolin at the cell protrusions enhances the local translation of CDH2 and thus increases cell invasive ability.

In conclusion, the oncologic function of SNHG15 is partially attributed to its ability to interact with IMP1 and nucleolin. Interaction with IMP1 allows SNHG15 to be localized to the cell protrusions, while association with nucleolin permits the protein to be co-transported with SNHG15. The accumulation of nucleolin at the cell protrusions finally enhances CDH2 translation and impacts cell invasive potential.

## 4. Materials and Methods

### 4.1. Cell Lines and Culture Conditions

Human breast cancer cell lines BT-549, MDA-MB-231, T47D and MCF7 were obtained from the American Type Culture Collection (ATCC). MDA-MB-231 and T47D cells expressing IMP1-GFP or GFP were previously established [21]. Breast stable cell lines expressing SNHG15 were established by infection with a lentivirus expressing MS2-tagged SHNG15, as previously described [44]. Cells were cultured in Dulbecco’s modified Eagle’s medium (DMEM) supplemented with 10% fetal bovine serum (FBS), 100 U/mL penicillin and 100 μg/mL streptomycin at 37 °C in a humid environment with 5% CO_2_.

### 4.2. Reagents

Primary antibodies against IMP1, GAPDH and nucleolin (NCL) were purchased from Cell Signaling Co. (Danvers, MA, USA); CDH2 and CDH1 antibodies were purchased from Boster Biological Technology Co. (Wuhan, China). Secondary antibodies conjugated with horseradish peroxidase (HRP) were purchased from Santa Cruz Biotechnology (Dallas, TX, USA). siRNAs against SNGH15, IMP1 and CDH2 mRNAs were purchased from Gene Pharma (Suzhou, China). PCR primers used in the study were obtained from IGE Biotechnology (Guangzhou, China) and are listed in Appendix A.

### 4.3. Isolation of IMP1 mRNP Complexes and Identification of IMP1-Associated lncRNAs

Briefly, T47D cells were lysed in an ice-cold lysis buffer containing 10 mM HEPES, pH 7.8, 40 mM NaCl, 10 mM KCl, 0.5% NP-40, 0.5 μg/mL PMSF and 1× protease inhibitor mixture (Roche). Supernatants were obtained after high-speed centrifugation (21,000× *g* for 60 min at 4 °C) and were incubated with IMP1 antibody and Protein A agarose beads (Sigma, St. Louis, MO, USA) at 4 °C in the presence of RNase inhibitor (250 units/mL). After overnight incubation, the supernatant was removed by a short centrifugation and the beads were extensively washed in lysis buffer followed by adding 1 mL of TRIzol. RNAs were extracted and sent to Shanghai Biotechnology Corporation (Shanghai, China) for RNA-seq assays. RNA-seq data have been deposited in the Gene Expression Omnibus under submission number GSE220087.

### 4.4. Cell Transfection, Lentivirus Assembly and Infection

Transfection of siRNAs and plasmids was conducted using Lipofectamine™ 8000 transfection reagent (Invitrogen, Waltham, MA, USA) following the protocol recommended by the manufacturer. At 48 h after transfection, cells were collected and used for further investigations. Lentivirus was generated by co-transfecting 293 T cells with the lentiviral vector and packaging plasmids as previously described [19]. Cultured supernatants were collected 48 h later, filtered through 0.45 μm filters and concentrated using Lenti-X concentrator (Clontech, Mountain View, CA, USA). Concentrated virus was used to infect cells immediately. Stably infected cell lines were selected with 2–5 μg/mL puromycin for about 2 weeks. The expression levels of expressed genes were detected by RT-qPCR.

### 4.5. Fluorescence in Situ Hybridization (FISH)

Breast cancer cells were grown on coverslips and were fixed with 4% paraformaldehyde (PFA) in diethylpyrocarbonate-treated PBS. FISH experiments were essentially performed as described previously [21]. FISH probes for detecting endogenous SNHG15, MS2-tagged SNHG15 and CDH2 mRNA are provided in Appendix A. All probes used in this study were synthesized and cy3-labeled by Gene Pharma (Suzhou, China).

### 4.6. Immunofluorescence (IF) Staining

Briefly, breast cancer cells were seeded on coverslips, fixed with 4% paraformaldehyde and pre-hybridized with 0.5% Triton X-100. Afterwards, cells were blocked and incubated with primary antibodies (1:200) at 4 °C overnight. The primary antibodies used in the experiments were anti-nucleolin, anti-CDH2 and anti-CDH1. After incubation, the coverslips were washed with PBS 3–5 times and incubated with fluorescent-labeled secondary antibodies at 25 °C for 1 h. The cells were then incubated with DAPI for 5 min at 25 °C for nuclear counterstaining. Images were captured using a confocal microscope (Zeiss, Munich, Germany).

### 4.7. Invasion and Proliferation Assays

Cell invasion assays were performed using Transwell chambers (Corning, New York, NY, USA). Briefly, cells (2 × 10^4^) were suspended in 200 μL DMEM containing 1% FBS and added to the upper chamber. Then, 700 μL medium containing 10% FBS was added to the lower chamber. After incubation for 20 h at 37 °C, cells on the upper surface were removed with a cotton swab. Cells on the lower surface of the membrane were fixed with 4% paraformaldehyde for 15 min and stained with 0.2% crystal violet for 10 min. The number of invasive cells was calculated for the entire lower surface. The experiment was repeated three times in triplicate. Cell proliferation assays were determined by using an MTT Cytotoxicity Assay Kit. Briefly, cells were seeded in 96-well plates at 5 × 10^3^ cells per well in a final volume of 100 μL. Cells were incubated for 2, 24, 48 or 72 h at 37 °C. After adding 10 μL (5 mg/mL) of MTT solution for 4 h, the supernatant was discarded and 150 μL of DMSO was added. Absorbance at 570 nm was measured using a microplate spectrophotometer (BioRad, Hercules, CA, USA). Each experiment was performed in triplicate and repeated three times.

### 4.8. Plasmid Construction

Human SNHG15 lncRNA was amplified by RT-PCR from total RNA extracted from MDA-MB-231 cells using the primers listed in Appendix A. Amplified SHNG15 was cloned into the pCIP2 lentivirus plasmid at the Not I and Bam HI sites. A plasmid expressing SNHG15-MS2_6_ was constructed by introducing a six-repeat MS2 hairpin structure to the BamHI site of the pCIP2-SHNG15 plasmid. Deletion of the IMP1 binding motif “ACACCC” from the SNHG15 sequence (Mut SNHG15) was performed using a Q5^®^ Site-Directed Mutagenesis Kit (NEB, Ipswich, MA, USA). The luciferase reporter PsiCHECK 2 (Promega, Madison, WI, USA) was used for luciferase activity assays. To construct luciferase reporter genes, the DNA fragments of the entire or truncated 3′UTR of CDH2 mRNA was cloned to the 3′ renilla luciferase gene of the PsiCHECK 2 plasmid.

### 4.9. RNA Isolation, qPCR and Western Blotting

RNA isolation, quantitative real-time polymerase chain reaction (qPCR) and Western blotting were performed as previously described [44]. Briefly, total RNA was isolated using TRIzol Reagent (Takara, Shiga, Japan). A total of 1 μg RNA was reverse transcribed to cDNA using a PrimerScript RT-PCR kit (Takara). The qPCR was conducted using a SYBR Green reaction mix (Vazyme, Nanjing, China). Relative expression was calculated using the 2^−ΔΔCt^ method (Ct, cycle threshold). All specific primers are listed in Appendix A. For Western blotting, cells were lysed in the presence of protease and nuclease inhibitors. Protein concentration was measured using a Pierce BCA Protein Assay Kit (Shanghai Branch, China). Equal amounts of proteins were electrophoresed using 4–12% SDS-PAGE and then transferred to nitrocellulose membranes (Merck, Rahway, NJ, USA), blocked and incubated with primary antibodies at 4 °C overnight. Primary antibodies used in blotting were anti-nucleolin, anti-CDH2, anti-CDH1 and anti-GPDH. After extensive washes, membranes were incubated with HRP-conjugated secondary antibodies at 25 °C for 1 h. Protein bands were visualized using an ECL kit (Invitrogen, USA). Where indicated, band intensities were determined by densitometry or using NIH ImageJ software (https://imagej.net/ij/index.html, accessed on 23 August 2023). For sequential immunoblotting experiments, the membranes were washed with Tris-buffered saline and treated with Western Blot Stripping Buffer (Thermo Scientific, Waltham, MA, USA) for 1 h. After washing and re-blocking, the membranes were incubated with other primary antibodies if necessary.

### 4.10. Protein Sequencing after MS2 Pull-Down Assays

MS2 pull-down assays were performed as previously described using a recombinant fusion protein (MBP–MCP) that contains a maltose-binding domain (MBP) and a domain (MCP) that recognizes the MS2 hairpins [44]. Briefly, recombinant MBP–MCP-conjugated amylose resin (NEB, USA) was incubated with cell lysates prepared from cells expressing MS2-tagged full-length or truncated SNHG15 at 4 °C for 4 h in the presence of RNase and protease inhibitors. After extensive washing, bound SNHG15-MS2 RNP complexes were eluted with 100 μL lysis buffer containing 20 mM maltose. Aliquots of the eluted materials were used for analyzing the enrichment of precipitated SNHG15-MS2 by RT-PCR. The remaining precipitates, which contained the proteins associated with SNHG15, were analyzed via spectrometry by Wininnovate Bio. Corporation (Shenzhen, China). Data of the protein spectrometry were submitted to the ProtemeXchange database with the submission number IPX0005630000.

### 4.11. Statistical Analysis

For statistical analysis, data from three independent qPCR results were calculated by the 2^−ΔΔCt^ method and represented as the means ± S.D. *p* values were determined using Student’s *t* test. Only *p* values lower than 0.05 were considered significant. * *p* < 0.05, ** *p* < 0.01 and *** *p* < 0.001.

## Figures and Tables

**Figure 1 ijms-24-15600-f001:**
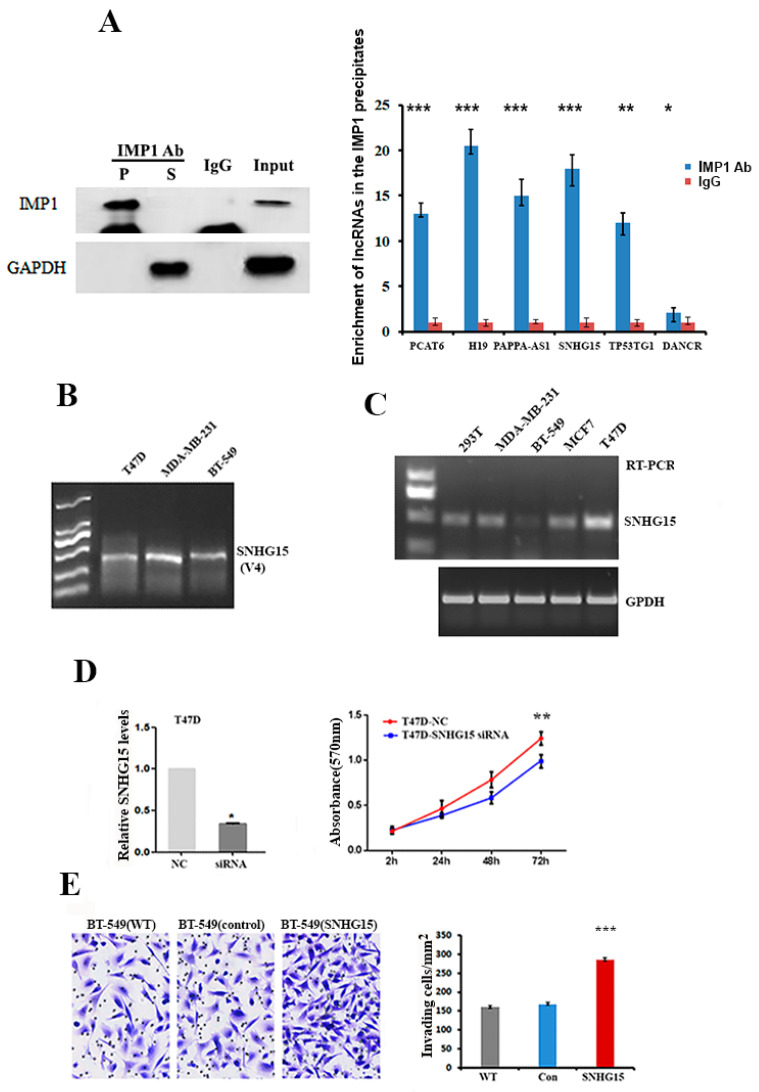
SNHG15 promotes the proliferation and invasion of breast cancer cells. (**A**) IMP1 antibody was used to perform RIP assays in BT-549 cells. Western blots (left panel) and RT-qPCR (right panel) show that IMP1 was truly associated with SNHG15 and five other selected lncRNAs. *** *p* < 0.001, ** *p* < 0.01 and * *p* < 0.05. (**B**) RT-PCR and gel electrophoresis show that the 983 bp (V4) SHNG15 is a major transcript in breast cancer cells. (**C**) RT-PCR and gel electrophoresis indicate the relative levels of endogenous SNHG15 in four breast cancer cell lines. GADP mRNA was used as an internal control. (**D**) Cell proliferation assays were performed in T47D cells, which showed that knocking down SNHG15 expression decreased cell growth potential. ** *p* < 0.01. (**E**) Transwell assays indicated that over-expression of SNHG15 in BT-549 cells increased cell invasive ability. *** *p* < 0.001.

**Figure 2 ijms-24-15600-f002:**
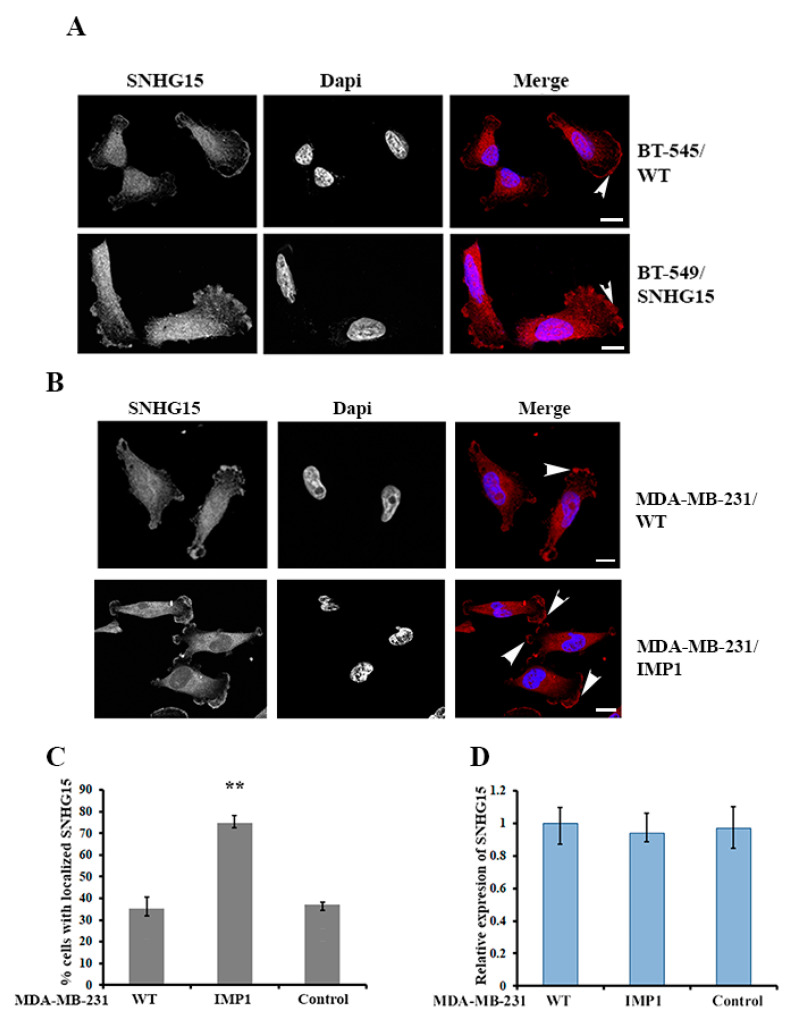
IMP1 regulates the localization of SNHG15 at the cell protrusions. (**A**) FISH was performed to detect the subcellular localization of SNHG15 in WT BT-549 cells (upper panel) and SNHG15-expressing BT-549 cells (lower panel). The arrowheads indicate localized SNHG15 at the cell protrusions. Scale bar: 10 µm. (**B**) FISH indicated that, in comparison with WT MDA-MB-231 cells (upper panel), IMP1 expression greatly increased the localization of SHNG15 at cell protrusions (lower panel). Scale bar: 10 µm. (**C**) A bar graph indicates the percentage of protrusion-localized SNHG15 in tested cells. Localization was increased to 70% from 40% when ectopic IMP1 was expressed. About 80–100 cells were counted in each group. ** *p* < 0.01. WT: cell origin. IMP1: cells expressing ectopic IMP1. Control: cells transfected with an empty plasmid. (**D**) RT-qPCR showed that IMP1 expression does not affect cellular levels of SNHG15.

**Figure 3 ijms-24-15600-f003:**
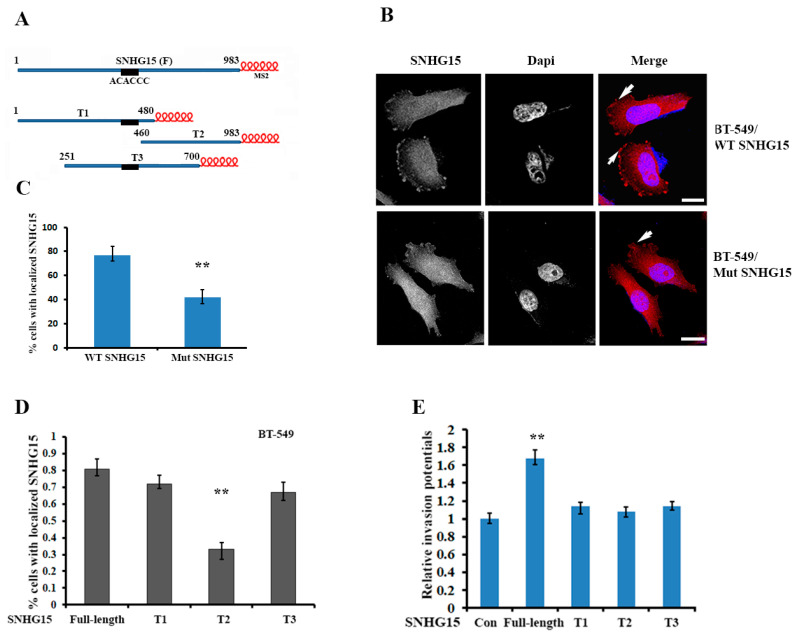
Localization of SNHG15 at the cell protrusions increases cell invasive potential. (**A**) The full-length SNHG15 and three dissected fragments of SNHG15 are separately tagged with six MS2 repeats. The relative position of the truncated SNHG15 is shown. The black box indicates the “ACACCC” motif for IMP1 binding. (**B**) FISH experiments showed that deletion of the “ACACCC” motif (Mut SNHG15) prevented SNHG15 from localizing at the cell protrusions. The arrowhead indicates cell-protrusion-localized SNHG15. Scale bar: 10 µm. (**C**) A bar graph shows that the cell population with protrusion-localized SNHG15 was largely decreased when the “ACACCC” motif was absent. ** *p* < 0.01. (**D**) Localization of the full-length and three truncated fragments of SNHG15 was determined by FISH. Results indicated that T1 and T3, but not T2, were still able to localize at the cell protrusions. About 80–100 cells were counted in each group. ** *p* < 0.01. (**E**) Transwell assays showed that, compared to the cells expressing full-length SNHG15, all three stable cell lines expressing truncated SNHG15 displayed lower invasive potentials. ** *p* < 0.01.

**Figure 4 ijms-24-15600-f004:**
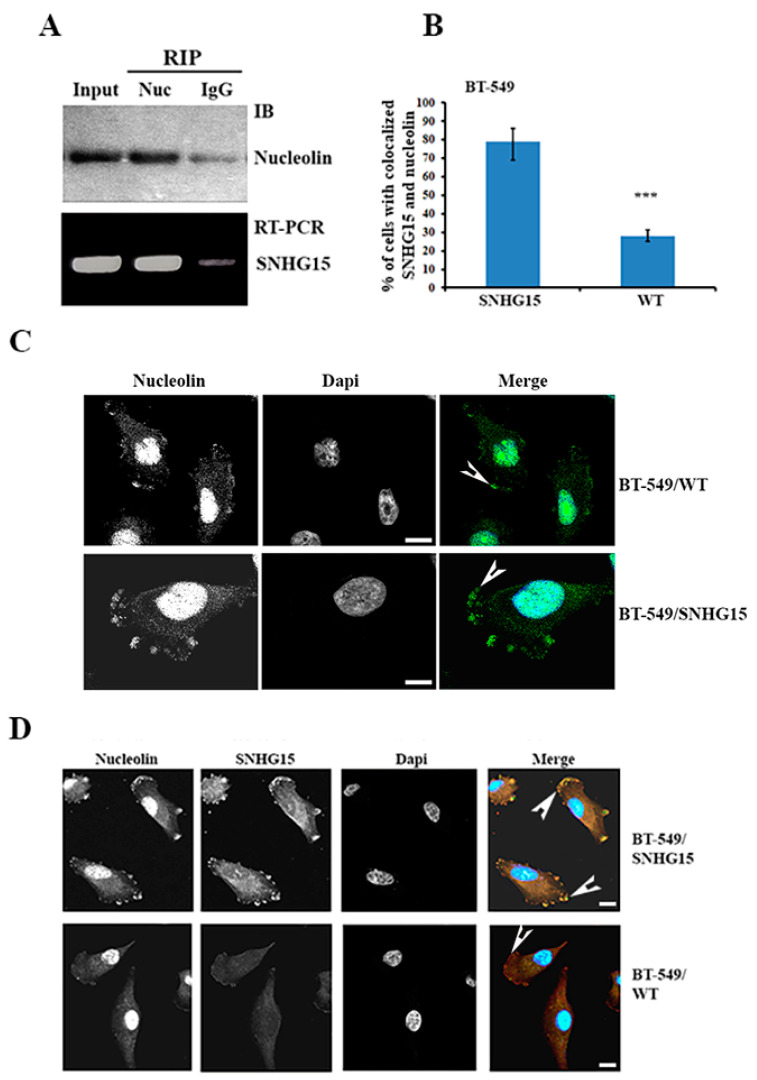
SNHG15 forms a complex with nucleolin and carries nucleolin to the cell protrusions. (**A**) RIP experiments using antibodies against human nucleolin were performed. Western blots and RT-PCR indicated that SNHG15 forms a complex with nucleolin. (**B**) A bar graph of IF assays indicates that accumulated nucleolin at the protrusions was greatly increased in SNHG15-expressing cells. *** *p* < 0.001. (**C**) IF assays were performed in BT-549 WT cells and cells expressing SNHG15 using nucleolin antibody. The accumulation of nucleolin at the cell protrusions was strongly increased in SNHG15-expressing cells. The arrowheads indicate protrusion localized nucleolin. (**D**) FISH and IF double staining assays showed that SNHG15 and nucleolin were co-localized (arrowheads indicated) at the cell protrusions of BT-549 cells. Scale bar: 10 µm.

**Figure 5 ijms-24-15600-f005:**
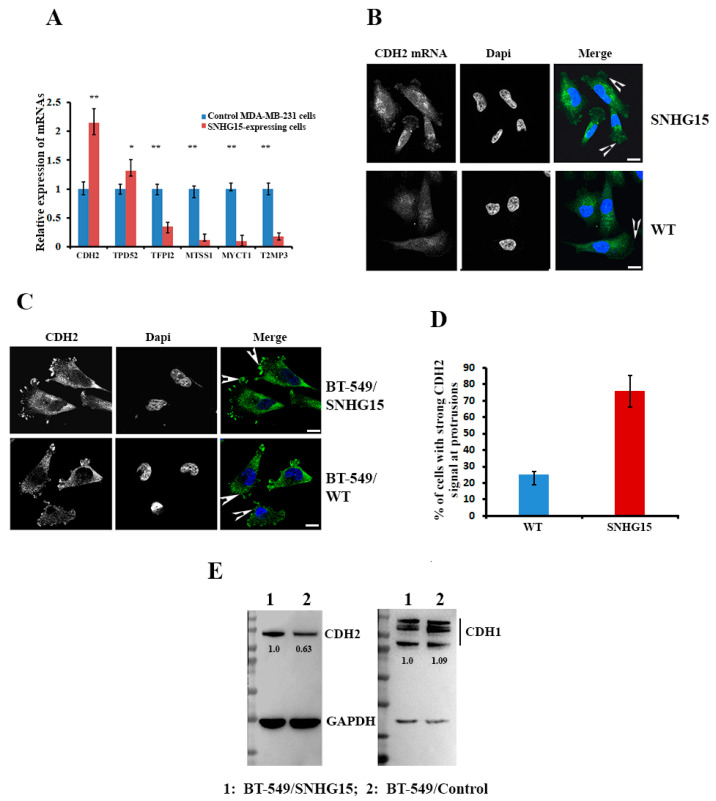
Expression of CDH2 mRNA was altered in responding to SNHG15 expression. (**A**) Six individual transcripts were selected to confirm their expression in response to SNHG15 expression in BT-549 cell lines. qRT-PCR showed that all selected mRNAs, including CDH2 mRNA, displayed a similar differential expression pattern, as indicated in BT-549 cells. ** *p* < 0.01, * *p* < 0.1. (**B**) FISH showed that substantial CDH2 mRNA was localized at the cell protrusion in SNHG15-expressing cells. The arrowheads indicate CDH2 mRNAs. Scale bar: 10 µm. (**C**,**D**) IF indicated that about 70% of cells showed CDH2 protein that was predominantly accumulated at the cell protrusions when SNHG15 was expressed. The arrowheads indicate detected CDH2 protein. Scale bar: 10 µm. (**E**) SNHG15 increased CDH2 but not CDH1 expression. Immunoblot analysis of proteins isolated from BT-549 cells with or without SNHG15 expression was performed. Numbers below the bands indicate relative levels of CDH2 and CDH1 proteins, which were normalized to GADH protein.

**Figure 6 ijms-24-15600-f006:**
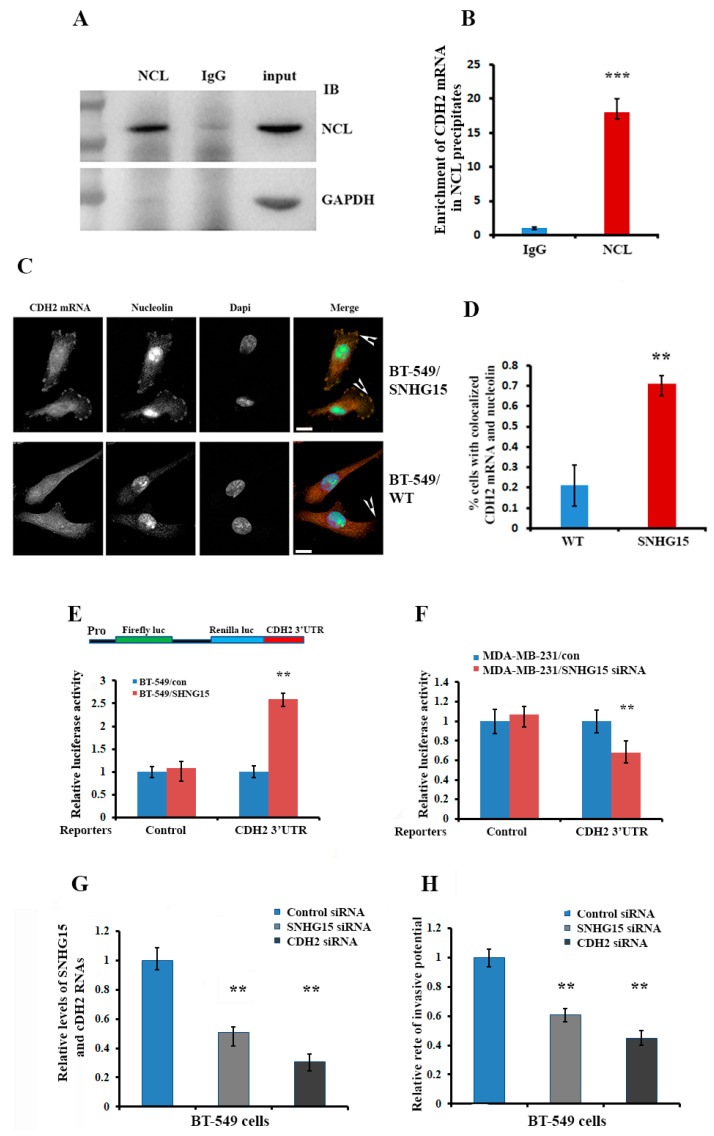
Accumulation of nucleolin at the cell protrusion enhances local translation of CDH2 mRNA and increases cell invasive potential. (**A**) Nucleolin antibody was used for immunoprecipitation of nucleolin and its associated RNA targets in breast cancer cells. Normal IgG and GAPDH were used as negative controls. *** *p* < 0.001. (**B**) CDH2 mRNA was preferentially bound to nucleolin. (**C**,**D**) IF and FISH double staining indicated that, in SNHG15-expressing cells, about 70% of cells showed strong co-localization of CDH2 mRNA and nucleolin at the cell protrusions (Arrowheads indicated). Scale bar: 10 µm. (**E**) Upper: the 3′UTR of CDH2 mRNA was cloned into the downstream region of the renilla luciferase gene of the PsiCHECK 2 dual luciferase reporter; lower: the reporter was transfected into SNHG15-expressing BT-549 cells for 48 h. Luciferase activity increased in cells expressing SNHG15. (**F**) The reporter was transfected into SNHG15-silenced MDA-MB-231 cells for 48 h. Luciferase activity was reduced when SNHG15 was knocked down by siRNA. ** *p* < 0.01. (**G**) Transwell assays were performed in SNHG15-expressing BT-549 cells in which SNHG15 or CDH2 mRNA was knocked down by corresponding siRNAs. ** *p* < 0.01. (**H**) Transwell assays were performed in MDA-MB-231 cells in which endogenous SNHG15 was knocked down by siRNA. ** *p* < 0.01.

## Data Availability

All data generated or analyzed during this study are included in this published article and its Appendix A.

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
