# Peer review of "SNHG15-Mediated Localization of Nucleolin at the Cell Protrusions Regulates CDH2 mRNA Expression and Cell Invasion"

_ijms, 2023, doi:10.3390/ijms242115600_

Round 1
Reviewer 1 Report
Chen et al., investigated the role lncRNA SNHG15 in mediating Nucleolin localization at the cell protrusions to regulate CDH2 expression. The authors demonstrated that SNHG15 recruits Nucleolin to the protrusion in BT-549 breast cancer cell, leading to an increase in CDH2 expression. To support their findings, they utilized a combination of various cell biology assays and omics techniques to unveil the oncogenic role of SNHG15 and the underlying mechanisms. While their findings are significant for cancer biology fields, a more extensive presentation of data would enhance clarity and comprehensiveness.
Major points
They need to show the overall analysis of their sequencing (by graphs, chart, etc e.g. for RIP-seq, show a graph to summarize the pulled down RNAs – class, length, expression level, GO terms).
Move the Fig S1a to the main figure. At least the qPCR graph. This is an important validation of RIP-seq.
The authors showed different expression levels of SNHG15 in different breast cancer cell lines. However, they used the BT-549 cell line throughout the manuscript, which has the lowest expression of SNHG15. They need to address why they chose BT-549 over others. Also need to be cautious about the gain of function effects by expression of SNHG15, considering the endogenous level of SNHG15 is very low.
Fig 1C and D, the authors need to clarify how to quantitate the invading cells for the readers with less background knowledge in this field
Throughout the figure sets, I highly recommend marking the protrusion regions with arrows or drawing. Also, they need to co-stain with a protrusion marker
In Fig3. They should have generated a full length SNHG15 construct without the ACACCC motif. Also either T1 with vs without motif or T2 with vs without motif should be tested to confirm the ACACCC motif is required for interaction
In Fig 4A. show MBP pulldown-Western blot with Nucleolin vs control antibodies
The author speculated that 3’ end of SNHG15 may be the Nucleolin binding site based on DNA sequencing analysis. However, can they confirm it by full-length vs truncated construction expression from Fig 3A? In discussion, they need to address the link between protrusion and SNHG15 mediated binding of Nucleolin (e.g. testing invasion by expressing ACACCC+T2)
In Fig 5. The authors need to show RIP-seq analysis. Show genome browser views of the target transcripts, and they can also quantitate the DEGs based on the reads
In Fig 5. They also need to show the comparison of their RNA profiles (SNHG15 induced) vs BT549 RNA profiles under normal invasive condition from others (you can check on GEO). Discuss how different they are, which could provide an insight into the SNHG15 mediated pathways
Minor points
In Fig 2. show IMP1 staining to show co-localization
In session 2.3. clarify MS2-MCP is used for future pull down. It is confusing whey MS2 was added until reading the next session
In Fig 4B, how about other breast cancer cell lines that express stronger SNHG15? Do they have endogenously stronger expression of Nucleolin than BT-549?
Fig 5e. it is hard to see the increase of CDH2. Quantification is needed.
In Fig 6. In order to conclude that SNHG15-mediated Nucleolin promotes CDH2 local ‘translation’, they need to show RNA level is stable just to make sure it is not through inhibition of post transcriptional silencing.
In 4.10. typo MS2-tagged floor-length à full-length
In 2.4. typo MBP-MBP à MBP-MCP
Author Response
Response to the Reviewer’s Comments:
Thank you very much for reviewing our manuscript. The followings please find our response to your excellent comments and suggestions.
Comments and Suggestions for Authors
Chen et al., investigated the role lncRNA SNHG15 in mediating Nucleolin localization at the cell protrusions to regulate CDH2 expression. The authors demonstrated that SNHG15 recruits Nucleolin to the protrusion in BT-549 breast cancer cell, leading to an increase in CDH2 expression. To support their findings, they utilized a combination of various cell biology assays and omics techniques to unveil the oncogenic role of SNHG15 and the underlying mechanisms. While their findings are significant for cancer biology fields, a more extensive presentation of data would enhance clarity and comprehensiveness.
Major points
They need to show the overall analysis of their sequencing (by graphs, chart, etc e.g. for RIP-seq, show a graph to summarize the pulled down RNAs – class, length, expression level, GO terms).
Based on the comment, we have provided a volcano plot as Suppl Fig S5A for RIP-seq analysis. We have also provided a table of interested genes from the RNA-sequencing as Suppl Fig S5B. We have also modified manuscript and mentioned analysis results.
Move the Fig S1a to the main figure. At least the qPCR graph. This is an important validation of RIP-seq.
We have removed Suppl Fig S1A to the Fig. 1 and modified the figure.
The authors showed different expression levels of SNHG15 in different breast cancer cell lines. However, they used the BT-549 cell line throughout the manuscript, which has the lowest expression of SNHG15. They need to address why they chose BT-549 over others. Also need to be cautious about the gain of function effects by expression of SNHG15, considering the endogenous level of SNHG15 is very low.
We thank the comments of the Reviewer. In our study, we not only used BT-549, but also used MDA-MB-231 cells for most studies. The lower SNHG15-expressing BT-549 cells is a good model for gain of function study and can avoid the interference by endogenous SNHG15. However, some other cell lines which expressed relative higher levels of SNHG15, such as T47D and MDA-MB-231 cells, to study the loss of function effects by silencing SNHG15.
Fig 1C and D, the authors need to clarify how to quantitate the invading cells for the readers with less background knowledge in this field
Based on the comment, we have added the methods for invasion and proliferation assays in the “method section” of the revised manuscript.
Throughout the figure sets, I highly recommend marking the protrusion regions with arrows or drawing. Also, they need to co-stain with a protrusion marker
In individual cells, protrusions are outward extensions of the plasma membrane, which adhere to the surrounding ECM and modulate cell migration. Many biological molecules including mRNA, protein and lncRNA could localize at this region to affect cell migration potential. Based on the suggestion, we have used arrows to indicate the localization of SNHG15, nucleolin and CDH2 mRNA at the cell protrusion regions. In addition, we have used IMP1 as a cell protrusion marker to show the colocalization of IMP1 with SNHG15 (Suppl Fig. S2A).
In Fig3. They should have generated a full length SNHG15 construct without the ACACCC motif. Also either T1 with vs without motif or T2 with vs without motif should be tested to confirm the ACACCC motif is required for interaction
We apologize for not adequately describing SNHG15 mutant construct and presenting the results in Fig 3. Actually, we have generated “a full length SNHG15 mutant without the ACACCC motif” (labeled as Mut SNHG15 in Fig. 3B and 3C, which were also described in the method section). We have shown that deletion of the motif notably reduced localization of SNHG15 at the cell protrusions. We have also used Mut SNHG15 and T2 (without motif) and wild-type SNHG15, T1 and T3 to teste localization function and invasion potentials, which are shown in Fig.3D and 3E. We have modified text in the revised manuscript.
In Fig 4A. show MBP pulldown-Western blot with Nucleolin vs control antibodies
The author speculated that 3’ end of SNHG15 may be the Nucleolin binding site based on DNA sequencing analysis. However, can they confirm it by full-length vs truncated construction expression from Fig 3A? In discussion, they need to address the link between protrusion and SNHG15 mediated binding of Nucleolin (e.g. testing invasion by expressing ACACCC+T2)
Based on the excellent comment of the Reviewer, we have confirmed that the 3’ end of SNHG15 bond to Nucleolin by performing nucleolin pulldown-RT-qPCR experiments. The results showed that truncated SNHG15 without the 3’end dose not bind to nucleolin. We have provided the result as Suppl. Fig S4C in the revised manuscript. we have also discussed this phenomenon and addressed importance of nucleolin-SNHG15 interaction for the protrusion localization and for cell invasive capability.
In Fig 5. The authors need to show RIP-seq analysis. Show genome browser views of the target transcripts, and they can also quantitate the DEGs based on the reads
In Fig 5. They also need to show the comparison of their RNA profiles (SNHG15 induced) vs BT549 RNA profiles under normal invasive condition from others (you can check on GEO). Discuss how different they are, which could provide an insight into the SNHG15 mediated pathways
We thank for the excellent comments of the Reviewer. After compare the RNA profiles between BT549/SHNG15 and BT549/control cells, we have modified text for the RIP-Seq analysis and provided a volcano plot and a table of gene list as Suppl Fig. S5A and S5B in the revised manuscript. We have briefly discussed the consequence of SNHG15 induced nucleolin localization for cell invasive potentials in the revised manuscript.
Minor points
In Fig 2. show IMP1 staining to show co-localization.
We have performed IF and FISH staining to show the colocalization of IMP1 and SNHG15 in BT-549 cells. The image has been provided as Suppl Fig. S2A in the revised manuscript.
In session 2.3. clarify MS2-MCP is used for future pull down. It is confusing whey MS2 was added until reading the next session
Thanks for the excellent suggestion. We have modified text and clarified that MS2-MCP would be used future for pull down assays.
In Fig 4B, how about other breast cancer cell lines that express stronger SNHG15? Do they have endogenously stronger expression of Nucleolin than BT-549?
Nucleolin is mainly a nuclear protein, but also plays functions in cytoplasm. In this study, we have also used MDA-MB-231 cell, where we found that nucleolin was also strongly colocalized with SHNG15 at the cell protrusions (Suppl Fig. S4B). Since MDA-MB-231 cells expressed relative higher endogenous SNHG15 than BT-549 cells, localization of nucleolin at the cell protrusions was also stronger in MDA-MB-231 cells than in WT BT-549 cells (Fig.6C and Suppl Fig. S4A)
Fig 5e. it is hard to see the increase of CDH2. Quantification is needed.
Based on the comment, we have performed densitometric analyses to quantitate the expression of CDH2. We have provided the quantitative results as Fig. 5F in the revised manuscript.
In Fig 6. In order to conclude that SNHG15-mediated Nucleolin promotes CDH2 local ‘translation’, they need to show RNA level is stable just to make sure it is not through inhibition of post transcriptional silencing.
In Fig. 6, we showed that CDH2 mRNA was preferentially co-precipitated with nucleolin (Fig 6A and 6B), while IF and FISH double staining assays indicated that nucleolin was strongly co-localized with CDH2 mRNA at the cell protrusions. We have also presented that SHNG15 expression increased CDH2 mRNA (Fig. 5A) and protein levels (Fig. 6E). Based on these data, we conclude that SNHG15-mediated nucleolin localization increased local CDH2 mRNA stability and translation.
In 4.10. typo MS2-tagged floor-length à full-length
In 2.4. typo MBP-MBP à MBP-MCP
We apologize for the typo errors. The errors have been corrected.
Reviewer 2 Report
The authors describe a novel interactions between lncRNAs SNHG15 with nucleolin and CDH2 mRNA and their involvement in the invasion of breast cancer cell lines. Although the manuscript is quite well written, several aspects need significant improvement prior manuscript acceptance.
Figure 2: The panel B depicting FISH experiments in cell lines with increased expression of IMP1 lacks arrows indicating SNHG15 localization. Also, what is the difference between WT and Control samples in panel C?
The authors claim that unlike full-length SNHG15, the truncated T1, T2 and T3 forms show lower invasive capability. And this phenomenon has not been discussed. Is it possible that this 6-MS2 hairpin loop repeat might negatively influence biological activity of SNHG15 and its truncated versions? Especially, that the interaction with nucleolin is through 3’ end of SNHG15. Or maybe truncated versions lack proper folding or 3D structure?
Line 177: I have an impression that the word “in vivo” has been misused. No animals were used in this experiment (and a whole manuscript), therefore all experiment were indeed “in vitro”
For RNA-seq analyses, a heatmap and/or volcano plot should be added to the figure 5, to show differentially expressed genes as a whole. Also, The list of all 223 differentially-expressed genes should be added to supplementary material, along with fold changes and FDR values.
Figure 5B: densitometric analyses should be performed for WB and added to the figure.
For all invasion transwell assay results, a representative images should be added.
Proliferation and invasion assays in 4.7 should be briefly described. The [44] reference does not describe these methods neither, and guides the reader to another paper, and so on…
The manuscript suffers moderate number of typos and grammar errors that need to be corrected by a native English speaker.
Author Response
Response to the Reviewer’s Comments:
Thank you very much for reviewing our manuscript. The followings please find our response to your excellent comments and suggestions.
The authors describe a novel interactions between lncRNAs SNHG15 with nucleolin and CDH2 mRNA and their involvement in the invasion of breast cancer cell lines. Although the manuscript is quite well written, several aspects need significant improvement prior manuscript acceptance.
Figure 2: The panel B depicting FISH experiments in cell lines with increased expression of IMP1 lacks arrows indicating SNHG15 localization. Also, what is the difference between WT and Control samples in panel C?
We have modified Fig 2B and put the arrows to indicate the localized SNHG15. We apologize for not describing clearly in Fig 2C. The “IMP1” indicates the cell line expressing ectopic IMP1 protein. The “control” indicates the cells transfected with an empty plasmid. “WT” indicates the cell origin. We have described the difference in the Fig.2 legend of the revised manuscript.
The authors claim that unlike full-length SNHG15, the truncated T1, T2 and T3 forms show lower invasive capability. And this phenomenon has not been discussed. Is it possible that this 6-MS2 hairpin loop repeat might negatively influence biological activity of SNHG15 and its truncated versions? Especially, that the interaction with nucleolin is through 3’ end of SNHG15. Or maybe truncated versions lack proper folding or 3D structure?
We have shown that the full-length SNHG15 simultaneously interact with IMP1 and nucleolin. Interaction with IMP1 allows SNHG15 to be properly localized, while interaction with nucleolin allowed SNHG15 to carry the protein to the cell protrusions. Thus, T1 or T3 which contains the IMP1 localization motif, but lacks the 3’end for nucleolin binding, while T2 fails to interact with IMP1, which explained why T1, T2 and T3 truncates unable to properly localize nucleolin at the cell protrusions and showed lower invasive capability. We have discussed this phenomenon in the revised manuscript.
In addition, the 6-MS2 hairpin loop has been used as an RNA-tag in many studies including our and other labs. The sequence of MS2 is well modified to prevent forming secondary structure, it is not likely that it can negatively influence biological activity of its tagged RNAs.
Line 177: I have an impression that the word “in vivo” has been misused. No animals were used in this experiment (and a whole manuscript), therefore all experiment were indeed “in vitro”
We thank the Reviewer for his excellent comment. The “in vivo” has been removed from the text.
For RNA-seq analyses, a heatmap and/or volcano plot should be added to the figure 5, to show differentially expressed genes as a whole. Also, The list of all 223 differentially-expressed genes should be added to supplementary material, along with fold changes and FDR values.
Based on the comment, we have provided a volcano plot for RNA-seq as Suppl Fig. S5A in the revised manuscript. We have also provided a table of gene list including CDH2, which were used for current and further study as Suppl Fig. S5B. In addition, we apologize for some errors in previous manuscript, containing description of “223 differentially-expressed genes”. We have modified text and corrected errors in the revised manuscript.
Figure 5B: densitometric analyses should be performed for WB and added to the figure.
Based on the comment, we have performed densitometric analyses for the WB of Fig. 5E. Relative levels of CDH2 and CDH1 proteins have been labeled below the protein bands which were normalized to GADH in the of Fig. 5F of the revised manuscript.
For all invasion transwell assay results, a representative images should be added.
We have added representative images for transwell assays as suppl fig. S3C and S6C in the revised manuscript.
Proliferation and invasion assays in 4.7 should be briefly described. The [44] reference does not describe these methods neither, and guides the reader to another paper, and so on…
We apologize for using incorrect reference [44] which misguides the reader. Based on reviewer’s comment, we have described the procedures for proliferation and invasion assays in the ‘method section’ and removed the reference in the revised manuscript.
Round 2
Reviewer 1 Report
I appreciate that the authors revised and improved all the comments I suggested.